# Design of High-Payload Ascorbyl Palmitate Nanosuspensions for Enhanced Skin Delivery

**DOI:** 10.3390/pharmaceutics16020171

**Published:** 2024-01-25

**Authors:** Jun-Soo Park, Jun-Hyuk Choi, Min-Yeong Joung, In-Gyu Yang, Yong-Seok Choi, Myung-Joo Kang, Myoung-Jin Ho

**Affiliations:** College of Pharmacy, Dankook University, 119 Dandae-ro, Dongnam-gu, Cheonan 31116, Republic of Korea; chon523273@gmail.com (J.-S.P.); wiiw10@naver.com (J.-H.C.); jmy951207@naver.com (M.-Y.J.); dhakflzk@naver.com (I.-G.Y.); analysc@dankook.ac.kr (Y.-S.C.)

**Keywords:** ascorbyl palmitate, nanosuspension, colloidal dispersion, hydrogel, bead-milling, dissolution, skin accumulation

## Abstract

A high-payload ascorbyl palmitate (AP) nanosuspension (NS) was designed to improve skin delivery following topical application. The AP-loaded NS systems were prepared using the bead-milling technique, and softly thickened into NS-loaded gel (NS-G) using hydrophilic polymers. The optimized NS-G system consisted of up to 75 mg/mL of AP, 0.5% *w*/*v* of polyoxyl-40 hydrogenated castor oil (Kolliphor^®^ RH40) as the suspending agent, and 1.0% *w*/*v* of sodium carboxymethyl cellulose (Na.CMC 700 K) as the thickening agent, in citrate buffer (pH 4.5). The NS-G system was embodied as follows: long and flaky nanocrystals, 493.2 nm in size, −48.7 mV in zeta potential, and 2.3 cP of viscosity with a shear rate of 100 s^−1^. Both NS and NS-G provided rapid dissolution of the poorly water-soluble antioxidant, which was comparable to that of the microemulsion gel (ME-G) containing AP in solubilized form. In an ex vivo skin absorption study using the Franz diffusion cell mounted on porcine skin, NS-G exhibited faster absorption in skin, providing approximately 4, 3, and 1.4 times larger accumulation than that of ME-G at 3, 6, and 12 h, respectively. Therefore, the high-payload NS makes it a promising platform for skin delivery of the lipid derivative of ascorbic acid.

## 1. Introduction

Ascorbyl palmitate (AP), a lipid derivative of ascorbic acid (Vit C), is widely used as an effective antioxidant in cosmetic or cosmeceutical formulations to protect the skin from oxidative stress and skin aging [1,2,3]. Lipophilic antioxidants show high potential to slow down the detrimental effects of photodamage [4]. To improve the skin absorption of AP, different dermal delivery systems and nanomaterials, such as microemulsions (MEs), solid lipid nanoparticles, liquid crystals, and liposomes, have been applied [3,5,6,7,8]. Although these pharmaceutical approaches are effective in increasing skin absorption and improving the chemical stability of labile antioxidants, the loading amount of AP in the topical delivery system is limited by its low solubility and/or loading in the vehicle [5,7]. The loading amount of AP in solid lipid nanoparticles, liquid crystalline, and liposomes were only 20 mg/mL, 25 mg/mL, and 12 mg/mL, respectively [6,7,8,9]. To the best of our knowledge, the ME system, which consists of 7.4 *w*/*v*% oil (Miglyol^®^ 812), 47.3 *w*/*v*% surfactant (Labrasol^®^ and Plurol^®^ oleique), and 45 *w*/*v*% aqueous solution [5], provides the highest AP content in the preparation (up to 5 *w*/*v*%) among the formulations reported to date. However, a colloidal system using a high proportion of surfactants (47.3 *w*/*w*%) may cause skin irritation upon repeated topical application [5,9].

Recently, drug nanosuspensions (NSs) have attracted much attention as viable tools for the topical and transdermal delivery of poorly water-soluble active compounds [10]. NS is a well-defined carrier-free colloidal delivery system composed of nanosized water-insoluble drug nanocrystals with a minimal quantity of suspending agent in the dispersion vehicles [11]. Particle size reduction to the nanoscale facilitates high drug loading in the preparation by dispersing the insoluble compounds in a solid state with low apparent viscosity [11]. Moreover, a decrease in particle size leads to an increase in the surface area, which is favorable for the adhesion of drug particles to biological membranes [12]. Moreover, the increased dissolution rate and drug concentration gradient between the stratum corneum and external preparations allow for higher penetration into the skin layer [13]. Nanocrystals < 500 nm even penetrate the skin in intact form via hair follicles or are absorbed by the surrounding follicular epithelium [14].

To design a stable NS delivery system, the synchronization of suspending and thickening agents that offer improved physical stability and the desired skin permeation profile is necessary. Amphiphilic and hydrophilic suspension agents are commonly used for this [11]. As amphiphilic materials, polysorbates 20 and 80, Kolliphor^®^ EL (Polyoxyl-35 castor oil), RH40 (Polyoxyl-40 hydrogenated castor oil), and Solutol^®^ HS15 (Polyethylene glycol 12-hydroxystearate) have been included in NS systems at concentrations of <2 *w*/*v*% [15]. These surfactants form a monomolecular layer on the surface of the drug particle, lowering the interfacial tension between the hydrophobic drug particles and the aqueous medium [16]. As hydrophilic polymers, cellulose derivatives such as carboxymethyl cellulose (CMC), hydroxypropylmethyl cellulose (HPMC), and hydroxypropyl cellulose (HPC), along with Polyvinylpyrrolidone (PVP), polyvinyl alcohol (PVA), and Carbopol^®^ (polyacrylic acid), are used to prevent the aggregation and precipitation of drug particles by interacting with and surrounding the surface of the drug particles [17]. These hydrophilic polymers with high molecular weight can also be employed as a thickening agent to adjust the viscosity of the external preparations [18,19]. However, incompatibility between the suspending and thickening agents occasionally cause the agglomeration and precipitation of dispersed drug particles [20]. In this regard, a sophisticated formulation design is required depending on the chemical structure or the crystalline form of the drug particles; however, to the best of our knowledge, no sophisticated formulation design and skin absorption profile of the NS system of the lipid derivative of ascorbic acid have been reported to date.

In this study, we aimed to design a sophisticated NS system for AP to increase its skin delivery. AP-loaded NSs were fabricated using a top-down bead-milling technique, and their physicochemical properties, such as morphology, particle size, surface charge, viscosity, and in vitro dissolution profile, were characterized. The soft hydrogel of NS (NS-G) was designed by evaluating the dispersibility of drug nanoparticles in the hydrogel matrix depending on thickening agents. Afterward, the skin absorption profile of AP after topical application of NS-G was comparatively evaluated with microemulsion-loaded hydrogel (ME-G) using an ex vivo Franz diffusion cell model mounted on porcine skin.

## 2. Materials and Methods

### 2.1. Materials

AP powder (purity over 98 *w*/*w*%), two grades of sodium carboxymethyl cellulose (Na.CMC) with an average molecular weight of ~90,000 and ~7,000,000 (named 90 K and 700 K), respectively, tyloxapol, Tween 80, xanthan gum, citric acid, and trifluoroacetic acid (TFA) were purchased from Sigma-Aldrich (St. Louis, MO, USA). Sodium lauryl sulfate (SLS) was obtained from Junsei Honsha Co., Ltd. (Tokyo, Japan). Polyvinylpyrrolidone K17 (PVP K17), macrogol (15) hydroxystearate (Kolliphor^®^ HS15), macrogolglycerol ricinoleate (Kolliphor^®^ ELP), macrogolglycerol hydroxystearate (Kolliphor^®^ RH40), polyoxyethylene (160) polyoxypropylene (30) glycol (Poloxamer^®^ 188), and carboxypolymethylene (Carbopol^®^ 934NF) were acquired from CTC Bio Inc (Gyeonggi, Republic of Korea). Hydroxyethyl cellulose (Natrosol™ 250, HEC), HPC E50 LV (Methocel™ E50), and HPMC (Methocel™ F4M) were kindly provided by Whawon Pharm. Co., Ltd. (Gyeonggi, Republic of Korea). Polyethylene glycol 4000 (PEG4000), capric triglyceride (Miglyol^®^ 812), caprylocaproyl polyoxyl (8) glycerides (Labrasol^®^), and polyglyceryl (3) dioleate (Plurol^®^ oleique) were kindly provided by Masung & Co., Ltd. (Seoul, Republic of Korea). Methanol and acetonitrile (ACN) were obtained from J.T. Baker (Phillipsburg, NJ, USA) and used without additional refinement.

### 2.2. Preparation of AP NSs and NS-Gs

AP-loaded NSs were prepared with a lab-scale wet-milling machine using dual centrifuging principles (Zentrimix^®^ 380 R; Andreas Hettich GmbH & Co KG, Tuttlingen, Germany), as previously described [21]. Approximately 5–15 mg of the suspending agent was dissolved in 1 mL of 50 mM citrate buffer (pH 4.5), and 25–75 mg of AP powder (average particle size of 5–10 μm) was subsequently added. Then, 1 g of zirconia beads (average diameter, 0.3 mm) was added and pre-wetted for 10 min using a multi-reax vortexer (Heidolph, Schwabach, Germany) at room temperature. The coarse suspension was bead-milled at different speeds of 1000, 1500, and 2000 rpm for 1 h at −10 °C. The prepared NSs were separated from the zirconia beads using 25G needle inserted syringes and stored in USP Type 1 borosilicate scintillation vials (Wheaton^®^, Millville, NJ, USA).

To prepare NS-Gs, several thickening agents, such as Carbopol^®^, Na.CMC 700 K, HEC, HPMC, and xanthan gum, were added to provide sufficient viscosity. Approximately 10–50 mg (1.0–5.0 *w*/*v*%) of the polymers was added to the prepared NSs and vortexed overnight using multi reax (Heidolph) at room temperature. All experiments were performed under light-shielded conditions, and the prepared formulations were stored in the glass scintillation vials at room temperature.

### 2.3. Preparation of AP-Loaded MEs and ME-Gs

For comparative evaluation, ME was prepared using the oil-in-water emulsification method, as previously described [5]. In a 4 mL glass vial, 74 mg of Miglyol^®^ 812 (an oil which has sufficient solubility for AP) was added, followed by the addition of 378 mg of Labrasol^®^ (a nonionic oil-in-water surfactant) and 95 mg of Plurol^®^ oleique (a nonionic hydrophobic surfactant) for emulsion formation [5]. After homogeneously dissolving all the components in the vial, 50 mg of AP was added and stirred at room temperature for 30 min to form a transparent solution. Then, 450 μL of distilled water was added and stirred vigorously for 3 h to form homogeneous emulsion droplets. All experiments were performed under light-shielded conditions. To prepare the ME-loaded soft hydrogel formulations (ME-Gs), Na.CMC 700 K was added at a concentration of 1.0% *w*/*v* and vortexed overnight using the same protocol described in Section 2.2.

### 2.4. Physicochemical Characteristics of AP-Loaded Nanoformulations

#### 2.4.1. Content Analysis of AP and Ascorbic Acid in Nanoformulations

AP and ascorbic acid concentrations in the formulations were analyzed using high-performance liquid chromatography (HPLC) as previously described [22], with slight modifications. Briefly, formulations corresponding to 50 mg of AP were fully dissolved in methanol at an AP concentration of 1.0 mg/mL in a 50 mL volumetric flask. The solution was then diluted five times with a diluent (0.1 *v*/*v*% TFA in ACN: distilled water = 90:10) and injected into the Shimadzu Prominence HPLC system composed of a pump (Model LC-20AD), a UV–VIS (ultraviolet-visible) detector (Model SPD-20A), and an autosampler (Model SIL-20AD) equipped with a C8 column (4.6 mm × 150 mm, 5 μm, Phenomenex^®^). The mobile phase, composed of 0.1% (*v*/*v*) TFA in ACN: distilled water = 90:10 (*v*/*v*), was eluted at a flow rate of 0.8 mL/min under isocratic conditions. The injection volume and column temperature were set at 20 µL and 4 °C, respectively. The eluent was monitored at 245 nm. The retention times of AP and ascorbic acid were approximately 2.1 and 4.7 min, respectively. The calibration curve for AP and ascorbic acid plotted between analyte concentration and the area under the peak was linear, yielding y = 23.83x + 21, R^2^ = 0.999 (50–200 μg/mL) and y = 61.47x + 24, R^2^ = 0.999 (25–200 μg/mL), respectively. The limit of quantification value was 20 and 10 μg/mL for AP and ascorbic acid, respectively.

To analyze the dissolved and suspended amounts of AP and ascorbic acid in NSs and NS-Gs, the supernatant was analyzed after centrifugation. The formulations were moved into a 2 mL sample tube and centrifuged at 13,000 rpm for 5 min. The supernatant was two-fold diluted with the mobile phase and analyzed using a Shimadzu HPLC system, as described above. The amount of AP suspended in the formulations was determined by subtracting the amount of AP dissolved in the supernatant from the total amount of AP.

#### 2.4.2. Particle Size and Surface Charge Analysis of Nanoformulations

The particle sizes of AP in NSs and NS-Gs and the oil droplet sizes in MEs and ME-Gs were measured using a Zetasizer Nano instrument (Malvern Instruments, Worcestershire, UK) at 25 °C [23]. Each sample was added to disposable cells after 60-fold dilution with distilled water. For each measurement, 20 runs were performed, and the data were processed as an average of three measurements. The zeta potentials of the nanosized AP in NSs and NS-Gs and oil droplets in MEs and ME-Gs were measured by loading 750 μL of 60-fold diluted samples onto a capillary zeta cell (DTS 1070; Malvern Instruments), and 20 repetitive runs were performed at 25 °C for each measurement [24]. The zeta potential was calculated as the average of three measurements.

#### 2.4.3. Stability Test under Stress Conditions

The physicochemical stability of an AP-loaded NS system depending on suspending agents was comparatively evaluated under stress conditions. At first, to evaluate the chemical stability of the AP-containing NSs, the formulations were aliquoted (2 mL per each aliquot) into 4 mL USP Type 1 borosilicate glass scintillation vials (Wheaton^®^, Millville, NJ, USA) and stored under 50 °C for 5 days. After 5 days of storage, the samples were cooled at room temperature, and the content of AP and AA was analyzed as described above.

The physical stability of the AP-containing NSs was assessed by means of a centrifugal stability test, as reported in a previous study [25]. The NSs were aliquoted (1 mL per aliquot, *n* = 3 for each formulation) into 1.5 mL polypropylene sample tubes and centrifuged at 13,000 rpm for 10 min. After centrifugation, samples were re-suspended using a multi-reax vortexer (Heidolph, Schwabach, Germany) for 1 min, and the particle size and homogeneity were evaluated as described in Section 2.4.2.

#### 2.4.4. Morphological Observation of Nanoformulations

The morphological features of the AP raw material were observed using scanning electron microscopy (SEM; Model JSM-6510; JEOL, Tokyo, Japan). AP powder was placed on an aluminum stub using double-sided tape and sputter-coated using an automatic sputter coater (Model 108AUTO; Cressington, UK) at 15 mA. Microphotographs of the coated samples were acquired at an acceleration voltage of 10 kV. Morphologies of the nanosized AP in NSs and NS-Gs and oil droplets in MEs and ME-Gs were observed using transmission electron microscopy (TEM; Tecnai F20 G2; FEI, Hillsboro, OR, USA). Approximately 4 μL of the formulations was loaded onto the copper grid and blotted for 1.5 s at 4 °C. Subsequently, the samples were fixed with liquid ethane using the plunge-freezing method and observed at an accelerating voltage of 200 kV.

#### 2.4.5. X-ray Diffraction (XRD) Analysis of Nanoformulations

The X-ray diffraction (XRD) patterns of the formulations were recorded on an X-ray diffractometer (Ultima IV; Rigaku Corporation, The Woodlands, TX, USA) using CuKα radiation with λ = 1.54 Å (40 kV and 35 mA) [26]. In the case of NSs and NS-Gs, to remove the excess vehicle components, the samples were centrifuged at 13,000 rpm for 10 min, and the precipitates were dried at room temperature for 24 h under light-shielded conditions. The dried formulations were then placed on a flat aluminum sample holder and scanned from 5° to 60° with a step size of 0.02° at a scanning speed of 2 s/step.

#### 2.4.6. Differential Scanning Calorimetry (DSC) Analysis of Nanoformulations

The thermal behavior of the formulations, including AP powder, NS, NS-G, and ME-G, was evaluated using DSC (DSC 50; Shimadzu Scientific Instruments, Tokyo, Japan), as previously reported [27]. The formulations were pre-treated as described above. Approximately 2 mg of the dried formulation was weighed, placed in a standard aluminum pan, and sealed with a lid. The heat flow associated with material transitions of each sample was recorded at a heating rate of 10 °C/min with a nitrogen purge of 20 mL/min. An empty aluminum pan was used as a reference.

#### 2.4.7. Fourier Transform Infrared (FTIR) Analysis of Nanoformulations

The crystallinity of AP in the nanoformulation was further evaluated using an FTIR spectrophotometer (Vertex-70V, Bruker Optics, Billerica, MA, USA) [28]. Drug powder, ME-G, and desiccated NS and NS-G samples (approximately 10 mg) were loaded in a horizontal attenuated total reflectance (ATR) accessory (Shelton, CT, USA) with a zinc selenide prism. FTIR spectra of samples were acquired using 32 scans with a resolution of 4 cm^−1^, with a spectral range from 600 to 2000 cm^−1^. The obtained spectra were processed through an OPUS/Mentor^®^ Software interface (https://www.bruker.com/en/products-and-solutions/infrared-and-raman/opus-spectroscopy-software.html, accessed on 19 December 2023).

#### 2.4.8. Apparent Viscosities of NS, NS-G, and ME-G

The apparent viscosities of the NSs, NS-Gs, and ME-Gs were determined using a rotational rheometer (ARES-G2; TA Instruments Ltd., New Castle, DE, USA) equipped with parallel plates (40 mm diameter) [29]. Approximately 2 g of the sample was added to the lower plate, and the upper and lower plates were spaced 1 mm apart. The samples were then subjected to the desired shear stress (10–1000 s^−1^). The temperature was set at 25 °C with a temperature accuracy of ±0.1 °C. The tolerance for each measurement was set to 5%.

### 2.5. In Vitro Dissolution Profiles of AP from NS, NS-G, and ME-G

Next, the in vitro dissolution profiles of AP in NS, NS-G, and ME-G were compared under sink conditions [30]. To guarantee sink conditions during the test, 0.5 *w*/*v*% SLS was added to 10 mM phosphate-buffered saline (pH 7.4). Different formulations containing 25 mg of AP were immersed into 200 mL of dissolution medium maintained at 32 ± 0.5 °C and agitated at a speed of 600 rpm in a shaking incubator. At predetermined time intervals, 1 mL of the dissolution medium was withdrawn and centrifuged at 13,000 rpm for 10 min. The supernatant was diluted two-fold with the HPLC mobile phase and the AP content was analyzed using HPLC, as described in Section 2.4.1. An equivalent volume of fresh pre-warmed dissolution medium was added to the dissolution vessel to maintain a constant medium volume.

### 2.6. Ex Vivo Skin Absorption Profiles of AP from NS-G and ME-G

To evaluate the accumulation and permeability of AP in the skin, ex vivo skin accumulation tests were performed on the dorsal skin of pigs (Cronex Co., Ltd., Suwon-si, Republic of Korea) using a Franz diffusion cell apparatus. In the receptor compartment, the medium was buffered to pH 3.0 with 10 mM succinate buffer to stabilize AP and ascorbic acid, and 0.5 *w*/*v*% SLS was added to ensure sink conditions. To prevent skin disintegration and microbial contamination, 0.02 *w*/*v*% thiosulfate was added. In the donor compartment, 200 μL of NS-G and ME-G formulations were added, and the top of the donor cell was sealed with parafilm to prevent the evaporation and unnecessary hardening of the formulations. The receptor solution was withdrawn at predetermined sampling points of 3, 6, 12, and 24 h. After two-fold dilution with the HPLC mobile phase, AP and ascorbic acid concentrations were analyzed using HPLC, as described in Section 2.4.1. To analyze the accumulated amounts of AP and ascorbic acid, the porcine skins were collected at the same time points, and the surface of the skin was wiped with methanol to remove the unabsorbed, surface-attached AP particles. Thereafter, the skin was sliced into small pieces, submerged in 10 mL of methanol, and vigorously vortexed for 24 h at room temperature to extract the AP accumulated in the skin tissues. The samples were centrifuged at 13,000 rpm for 10 min to remove the skin fragments or undissolved matters. The supernatant was then diluted two-fold with the HPLC mobile phase for AP quantification using HPLC, as described above. The accumulated amount of AP was calculated by dividing the accumulated amount by the weight of the skin (μg/g) using Equation (1) [14]:(1)$$\frac{Concentration\;of\;AP\;in\;the\;supernatant\;(\mu g/mL)\times Volume\;of\;extraction\;solvent\;(mL)}{Weight\;of\;the\;porcine\;skin\;(g)}$$

### 2.7. Statistical Analysis

Each experiment was conducted at least three times, and the data were expressed as mean ± standard deviation. The significance of the data was statistically analyzed using Student’s *t*-test with a significance level of *p* < 0.05 unless otherwise specified.

## 3. Results and Discussion

### 3.1. Screening of Suspending Agents to Formulate AP-Loaded NSs

Here, we selected the NS drug delivery system to formulate the topical preparation of AP, a lipophilic antioxidant. Compared to conventional emulsion or oily formulas, the NS system has the advantages of a high drug payload, improved stability, and lower use of surfactant or organic solvent, which can cause skin irritability [31]. The NSs were fabricated using a wet-milling technique, a top-down process that relies on mechanical attrition to convert large crystalline particles into submicron particles [21]. The wet-milling technique employing beads as a collision catalyst was adopted to formulate an AP-loaded NS system. This simple and robust fabrication process offers several advantages, such as ease of scale-up, little to no yield loss, low batch-to-batch variability, and no use of harmful organic solvents. Moreover, the production costs of NSs are generally low because of the low excipient requirements during their preparation compared to conventional formulation such as ME, lipid nanoparticle, and liquid crystalline systems [23].

However, during the milling process, nanosized particles in the NS formulation tend to aggregate because of Ostwald ripening and the Brownian motion of drug particles [32]. To avoid agglomeration during and after the milling process and to stabilize the nanosized particles, appropriate suspending agents such as surfactants and/or polymers should be added during the preparation process. In this study, to select the appropriate suspending agent for the NSs, different surfactants and polymers were screened at a concentration of 10 mg/mL (1 *w*/*v*%) with a milling intensity of 1500 rpm for 1 h, and the AP concentration was 50 mg/mL (5 *w*/*v*%). The temperature of the milling vessel was set to −10 °C to prevent undesirable temperature elevation, which could accelerate the degradation of AP. The particle size and homogeneity of the AP nanoparticles in the NSs were evaluated, and the flowability and dispersibility were visually inspected (Table 1). When PEG 4000 or Na.CMC 90 K were employed as suspending agents at a concentration of 10 mg/mL, drug crystals larger than 2300 nm were fabricated. NSs prepared with tyloxapol and PVP K17 resulted in a viscous appearance with particle sizes of 582 and 772 nm, respectively. On the other hand, the preparation of NSs with surfactants including Tween 80, Kolliphor^®^ HS15, Kolliphor^®^ RH40, Kolliphor^®^ ELP, and Poloxamer^®^ 188 offered a flowable and homogeneous appearance with sub-micron particle size (<660 nm) with low PDI value (<0.31). However, in the case of NSs fabricated with Poloxamer^®^ 188 and PVP K17, the AP content in formulas decreased right after preparation to 46.5% and 53.4%, respectively, probably reflecting drug instability.

The physical stability of the NSs prepared with Tween 80, Kolliphor^®^ HS15, and RH40 was evaluated by means of a centrifugal stability test, as depicted in Section 2.4.3, to assess the degree of particle sedimentation or aggregation [25]. The size distribution and homogeneity were compared before and after centrifugation and depicted in Figure 1A. The particle sizes substantially increased in Kolliphor^®^ ELP and tyloxapol-containing NSs from 655.6 to 1338.0 nm and 582.1 to 2525.0 nm, which is more than a two-fold and four-fold increase, respectively. On the other hand, only slight increases in the particle size were observed in NSs fabricated with Tween 80, Kolliphor^®^ HS15, and RH40 after centrifugation, from 479.7, 479.3 and 488.1 nm to 579.3, 527.9, and 554.6 nm, respectively.

As AP is vulnerable to hydrolysis, the surrounding aqueous medium of NSs could accelerate its degradation, particularly with the increased surface area of the nanoparticles, as reported in a previous NS formulation study containing ginkgolides [33]. Therefore, we further evaluated the chemical stability of AP-loaded NSs fabricated with Tween 80, Kolliphor^®^ HS15, and RH40 under stress conditions. The AP contents right after preparation and after 5 days stored at 50 °C were depicted in Figure 1B. The AP content significantly decreased in NSs fabricated with Tween 80 and Kolliphor^®^ HS15 to 76.5% and 82.4%, respectively. However, more than 86% of AP was retained in Kolliphor^®^ RH40-containing NSs. Actually, AP NS prepared with Kolliphor^®^ RH40 was physicochemically stable under long-term storage conditions (25 °C/RH60% and 40 °C/RH75), with over 95% of the drug remaining after 4 weeks. In addition, the particle size of NS ranged between 450 and 500 nm after 4 weeks, with no particle aggregation. Therefore, Kolliphor^®^ RH40 was selected as an appropriate suspending agent for the AP NS formulation. Kolliphor^®^ RH40, a polyoxyl-40 hydrogenated castor oil, was also considered a suitable suspending agent from toxicological and environmental perspectives. It was reported that the surfactant is not acutely harmful to aquatic organisms, with complete biodegradation in water [34].

### 3.2. Effects of Formulation Variables on Particle Size and Homogeneity of NSs

The influence of the milling intensity, concentration of the suspending agent, and AP content on the particle size, homogeneity, and solubility of AP in formulations were depicted in Figure 2. At first, the effect of milling intensity on NS particle size was evaluated under three different suspension agent concentrations: 0.5, 1.0 and 1.5 *w*/*v*% (Figure 2A). At 1000 rpm of milling speed, the particle sizes were greater than 1000 nm for all the suspending agent concentrations. As the concentration of the suspending agent increased from 0.5 to 1.5 *w*/*v*%, the particle size increased slightly from 1088 to 1208 nm. It is assumed that excessive suspending agents can form a steric layer on the surface of the AP particles and increase the hydrodynamic diameter of the particles, as reported in previous reports of polyethylene glycol-modified polymeric particles [35]. On the other hand, with an increased milling intensity of 1500 rpm, AP particles were pulverized below 600 nm at all suspending agent concentrations, and the increased surface area of the nanosized AP particles was successfully stabilized. In contrast to the 1000 rpm milling speed, the AP particle size of AP was sequential as the concentration of the suspending agent increased, from 593.9 nm to 427.5 nm. At a milling speed of 2000 rpm, AP NSs smaller than 450 nm were obtained, and a slight decrease in the size of the AP NSs was observed as the concentration of the suspended agents increased, from 440.8 nm to 417.6 nm.

In pharmaceutical suspensions, lowering the dissolved amount of the active compound is desirable to prevent the recrystallization of active pharmaceutical ingredients, which can cause polymorphism in crystals and increase particle size [36]. In addition, ascorbic acid is vulnerable to aerobic and oxidative conditions that create dehydroascorbic acid, which is very unstable during hydrolysis [37]. Maintaining an undissolved state is favorable for its chemical stability. Therefore, the dissolved percentage of AP was analyzed using different concentrations of the suspending agent (Figure 2B). As the concentration of the suspending agents increases, from 0.5 to 1.5 *w*/*v*%, the dissolved concentration of AP was sequentially increased to 345.4, 694.5, and 1076.1 μg/mL, with percentages of 0.7, 1.4, and 2.2 *w*/*v*%, respectively. This linear increase in the dissolved amount of AP indicates that the suspension agent concentration was above the critical micelle concentration [38]. Because the homogeneity of the NSs was sufficiently secured with 0.5% *w*/*v* suspending agents, the suspending agent was fixed at 0.5% *w*/*v*.

The effect of AP concentration on particle size was evaluated with an optimized milling speed and suspending agent concentration of 2000 rpm and 0.5% *w*/*v*, respectively (Figure 2C). As the concentration of AP increases from 2.5 to 5.0 *w*/*v*%, the average particle size of NS decreased slightly from 467.1 to 431.1 nm. Moreover, there was no significant difference in the average particle size between the AP concentration of 5.0 and 7.5 *w*/*v*% NSs (428 and 430 nm). With only 0.5 *w*/*v*% of the suspending agent, which is substantially lower than the surfactant amount of ME formulation (47.3 *w*/*w*%), high-payload (up to 7.5 *w*/*v*%) AP-containing NSs were successfully fabricated. Additionally, the drug-loading amount in NS (up to 7.5 *w*/*v*%) is approximately 3.5, 3.0, and even 6.1 times higher than that of solid lipid nanoparticles, liquid crystalline, and liposomes previously reported, respectively [6,7,8,9]. It has been reported that low-molecular-weight lipophilic compounds with generally molecular weights of less than 400 Da permeate the stratum corneum principally through passive diffusion through the lipid bilayers by hopping between the free volume pockets [39]. Accordingly, the high loading of AP in the NS system was expected to promote skin absorption of AP through the diffusion pathway, with an increased concentration gradient between the external preparation and skin layer. For further comparative evaluation with the ME formulations, the AP content was set to 5.0 *w*/*v*% in NS formulation.

### 3.3. Design of NS- and ME-Loaded Soft Hydrogel Formulations (NS-Gs and ME-Gs)

For better skin retention and physical stability, the AP-containing NS was further embedded into soft hydrogel formulation (NS-G) by employing biocompatible thickening agents (xanthan gum, Carbopol^®^, HEC, HPMC, and Na.CMC 700 K). Although the increase of medium viscosity could increase the physical stability of the NS by lowering the sedimentation rate according to Stokes’ law [18], excessively high viscosity is unpleasant for skin application. In addition, in order to enable topical application with a spraying device as well as direct application, we aimed to design the NS-G with viscosity below 15 cP, as described in previous studies [40,41]. The homogeneity of drug nanocrystals in the hydrogel matrix depending on the thickening agent was scrutinized using an optical microscope (Figure 3). Aggregation of the thickening agent (whitish aggregation) or large-sized agglomerates of AP nanoparticles were observed in xanthan gum 1 *w*/*v*%, Carbopol^®^ 1 *w*/*v*%, HEC 5 *w*/*v*%, and HPMC 2 *w*/*v*%-based hydrogels. On the other hand, NS-G prepared with Na.CMC 700 K resulted in homogeneous gel networks without thickening agent agglomerates or AP particle aggregations (Figure 3C). In combination with Na.CMC 700 K and AP NSs, hydrogen bonding might be formed between the hydroxyl and carboxyl groups in the Na.CMC 700 K and AP molecules, as previously reported [42], stably embedding the drug nanoparticles. To fabricate ME-G, 0.5 *w*/*v*% Na-CMC was also employed. CMC is a fundamentally non-toxic, biodegradable polysaccharide, and is estimated to be degraded entirely in environmental systems [43], causing neither toxicological nor environmental issues.

### 3.4. Physicochemical Characteristics of Different Nanoformulations

The physicochemical characteristics of three different AP-containing formulations, NS, NS-G, and ME-G, were summarized in Table 2, Figure 4 and Figure 5. The morphologies of the raw materials NS and NS-G were scrutinized using SEM and TEM, as depicted in Figure 4A–C. The AP raw material showed long flaky morphologies with lengths of more than 10 μm of the long axis. As amphiphilic AP molecules with palmitate chains formed a crystal lattice in the longitudinal direction, long flaky crystals were formed. Although the particle sizes decreased to 500 nm in NS and NS-G, similar morphological characteristics of long and flaky particles were observed (Figure 4B,C). In contrast, in the case of the ME-Gs, uniform spherical oil droplets approximately 100 nm in diameter with no drug particles were observed under TEM examination (Figure 4D).

The total, suspended, and dissolved concentrations of AP in each formulation are summarized in Table 2. The total concentrations of NS, NS-G, and ME-G were in the range of 99.4–105.5% and the dissolved AP concentrations in NS and NS-G were less than 0.35 mg/mL, which is 0.7% of the total AP concentration. The characteristics of the nanoparticles, including particle size, emulsion droplet size, homogeneity, and surface charge, are summarized in Table 2 and depicted in Figure 5A. The size of the AP particle in NS and NS-G was similar with an average diameter of 489.5 and 493.2 nm, respectively (Figure 5A). The size of NS, NS-G, and ME-G formulations was homogeneous, with PDI values of <0.3. The surface of the nanoparticle was negatively charged in NS and NS-G, with −33.0 and −48.7 mV, respectively, which provides a desirable repulsive force between particles that prevent particle growth and aggregation [44]. The more negative surface charge of the NS-G was probably due to the adsorption of the anionic cellulose polymer [45]. In the case of ME-G, negatively charged polymers might have become attached to the surface of the oil droplets and showed a negative charge of −41.0 mV.

To confirm the changes in crystallinity in NS, NS-G, or ME-G, DSC thermograms, XRD patterns, and FTIR spectra of each formulation were evaluated (Figure 5B–D). The endothermic trough observed in AP raw material was decreased in NS formulation 118 °C to 114 °C. The pharmaceutical excipients, including Kolliphor^®^ RH40 minorly contained in the desiccated sample, which has a congealing range of 16–26 °C, might result in the partial shift of the endothermic peak [46]. In NS, an additional endothermic peak was observed around 80 °C (Figure 5B), which was not observed in the vehicle, denoting the formation of another crystalline form with weaker intermolecular bonding. In the XRD patterns, the distinctive, sharp XRD pattern of AP powder was mostly retained in the system in NS, preserving characteristic peaks at 7.5, 15.2, 19.9, 21.1, 22.9 and 25.6°. However, the XRD patterns around 16–25° were partially altered, with variations in peak intensity (Figure 5C). For the further evaluation of crystallinity, FTIR spectra of formulations were acquired to collect additional information on the structural modification of AP in NS, NS-G, and ME-G (Figure 5D). In the FTIR spectrum of AP powder, bands in the region of 1730, 1637, and 1467 cm^−1^ originating from C=O stretching of ester, double-bond C=C in ascorbic acid, and the CH_3_ group, respectively, were observed [47]. The absorptions in the regions around 1344 and 1159 cm^−1^ are interconnected to the C–O–C bond of AP [47]. The spectrum of NS was analogous to that of AP powder overall, but the absorption bands in 1637 cm^−1^ corresponding to the C=O functional group of ascorbic acid was observed to have shifted to 1656 cm^−1^. Moreover, the absorption bands in 1000–1400 cm^−1^ corresponding to the C–O–C bond of AP were broadened (Figure 5D). Taken together, we assumed that AP particles in the NS system principally retained their original crystalline form, but some fraction of the drug particles were converted into another crystalline form with a lower melting point during the bead-milling process. The addition of a thickening agent did not affect the crystallinity of the AP particles, with identical DSC, XRD, and FTIR patterns for the NS and NS-G formulations, respectively (Figure 5B–D). On the other hand, in the DSC thermogram of ME-G, no noticeable endothermic trough was observed (Figure 5B). In addition, no diffraction peak was observed in XRD analysis, collectively indicating that the lipid prodrug was molecularly incorporated into the oil droplets in solubilized form. FTIR analysis further supported the conclusion that AP was contained in ME-G in dissolved form, showing noticeably weakened and broadened absorption bands (Figure 5C).

The rheograms of NS, NS-Gs, and ME-Gs were represented in Figure 5E. The addition of Na.CMC 700 K polymer into NS (NS-G) increased the viscosity, providing > 10 times higher viscosity of between 1.6 and 3.5 cP. NS-G showed a typical pseudoplastic flow pattern with viscosity decreasing as shear rate increased. On the other hand, ME-G showed a Newtonian flow pattern with constant viscosity (7.5–7.6 cP) with a shear rate of 10–1000 s^−1^. The viscosity of ME-G was higher than that of NS-G with an equal amount of Na.CMC 700 K polymer, as ME itself is composed of high-viscosity oil, a co-solvent, and a surfactant with viscosity of 25–33 cP, 80–110 cP and 3000 cP, respectively [48,49,50]. The pH values of NS and NS-G were identically set to 4.5, which is within the recommended pH range for topical products of 4 to 6 [51].

### 3.5. In Vitro Dissolution Profiles of AP from Different Formulations

In vitro AP dissolution profiles of the raw material, NS, NS-G, and ME-G, were comparably evaluated under sink conditions (Figure 6). An anionic surfactant, 0.5 *w*/*v*% SLS, was added to PBS (pH 7.4) to provide sufficient AP water solubility (about 390 μg/mL), which is sufficient solubility for drug dissolution.

Under sink conditions, about 60% of the added drug powder was dissolved in 5 min followed by gradual drug release for 2 h (Figure 6). After the rapid dissolution of the drug powder under sink conditions, the remainder was aggregated to form drug clusters, which delayed the dissolution rate with diminished surface area. In contrast, the pulverization of AP powder into nanocrystals (NS) drastically promoted drug dissolution, exhibiting over 89.0% drug release after 5 min. Afterward, complete dissolution over 95% was observed in the NS system within 15 min. As the average particle size decreased to one-twentieth that of the raw material, the surface area of the AP particles substantially increased in NS. Thus, according to the Noyes-Whitney equation, an increase in the particle surface area results in a proportional increase in the dissolution rate [52]. The dissolution profile of NS-G was comparable to that of NS, denoting the fact that the soft hydrogel network did not hamper the penetration of the dissolution medium and drug release from the gel matrix, with low viscosity. In addition, after 10 min of dissolution, NS and NS-G resulted in over 90% dissolution, which is comparable to that of ME-G, which contains AP in a dissolved state inside the nanosized oil droplet. Therefore, we concluded that NS-G could provide rapid and profound dissolution comparable to ME-G, which would be beneficial for skin absorption of the lipophilic antioxidant.

### 3.6. Ex Vivo Skin Absorption Profiles of AP from NS-G and ME-G

In our study, skin accumulation and permeation profiles of AP were evaluated using a Franz diffusion cell mounted on porcine skin following the topical application of NS-G and ME-G formulations containing 50 mg/mL of AP. Porcine skin is widely used in ex vivo skin absorption studies because of its anatomical, physiological, and immunological similarities to human skin. The thickness of the stratum corneum (SC) of porcine skin is reportedly 20–26 μm, comparable to that of human skin [53]. Moreover, the lipid and protein compositions of the different skin layers are also analogous between both species [54], with a comparable dermal–epidermal thickness ratio (10:1–13:1) [55]. Regarding the blood supply in the dermis, porcine skin is even comparable to human skin [56]. Furthermore, considering its availability and low individual sample deviation, porcine skin is regarded as an alternative to human skin [14]. To guarantee sink conditions for drug permeation, 0.5 *w*/*v*% of SLS was included in PBS (pH 7.4).

The accumulation profiles of AP following the topical application of NS-G or ME-G are represented in Figure 7. In both formulations, the amount of AP deposited in the skin tissues increased proportionally throughout the experimental period. Interestingly, at an early time point (3 h), a markedly larger amount of AP was deposited in the skin after NS-G application (292.4 μg/mg), which is more than four times larger than that of ME-G (72.4 μg/mg). NS-G also provided higher drug accumulation after 6 and 12 h post-application, exhibiting approximately three and 1.4 times larger accumulation than that of ME-G, respectively. Afterwards, the amount of AP deposited in the skin after 24 h was comparable between NS-G and ME-G (1258–1311 μg/mg).

Interestingly, the NS system provided higher initial skin accumulation compared to the ME formula, despite a particle size approximately five times larger. This excellent skin absorption profile of NS might be attributed to its increased adhesiveness to the skin layer with a high dissolution rate. According to Dragicevic et al. (2016)’s report, a nanocrystal-loaded NS-G formulation exhibited increased adhesiveness to cell surfaces or membranes, which further increased the residence time of the formulation in the skin [57]. Subsequently, the increased saturation solubility in the skin increases the concentration gradient between the formulation and the skin, leading to a higher diffusive flux according to Fick’s first law [58]. Oktay et al. (2018) reported that the higher saturation solubility of the flurbiprofen-loaded NS system compared to that of the coarse powder resulted in an increased concentration gradient and permeation flux value [59]. In addition, intact drug nanocrystals below 500 nm can directly penetrate hair follicles and glandular structures, such as sweat and sebaceous glands, with no additional dissolution step [60]. Conversely, the skin absorption of AP after ME-G application was retarded compared to NS-G, probably delaying the time taken for the drug molecule to partition out of the ME into the skin layer [61]. As AP is a highly lipophilic compound with an estimated log *p* value of 6.0 [62], the lipidic drug resides favorably in the oil component (Miglyol^®^ 812) with high logP > 8 rather than in the skin layer [48].

On the other hand, neither AP nor ascorbic acid, a hydrolysate of AP, were detected in the receptor phase after NS-G or ME-G application throughout the experimental period (24 h). This result is consistent with previous findings that the lipophilic derivatives of ascorbic acid hardly penetrated skin tissue because of the low diffusion rate of longer-chain fatty acids [63]. Gosenca et al. (2011) also reported that the AP molecule did not permeate excised pig ear skin after the application of emulsion formulas [64]. Rather, AP accumulates in the epidermal and dermal layers of the skin, which can act as a very strong reservoir for lipophilic compounds [58]. On the other hand, AP was reported to be converted to ascorbic acid by esterase in the skin layer, and some fraction of ascorbic acid permeated the skin layer [7]. However, in this study, ascorbic acid remained in the skin or penetrated below the limit of quantification. In the presence of SLS added to the receptor phase to maintain sink conditions, the conversion of AP to ascorbic acid might be minimized because of the esterase-inhibitory effect of SLS, as previously reported [65]. The SLS disrupts the non-covalent bonds in enzymatic proteins and denatures them, causing the enzymes to lose their native conformations and shapes [65]. Carruthers et al. (1962) reported that esterase activity decreased to one-tenth in the presence of 0.1 *w*/*v*% SLS in rat liver microsomal homogenates [66]. Taken together, we concluded that the novel NS system is more effective in increasing the skin absorption of AP than the conventional ME system, with minimal use of surfactant.

## 4. Conclusions

In this study, a sophisticated NS system was lucratively prepared using bead-milling technology, employing Kolliphor^®^ RH40 (0.5 *w*/*v*%) and Na.CMC 700 K (1.0 *w*/*v*%) as suspending and thickening agents, respectively. The designed NS system was capable of dispersing high amounts of AP (up to 75 mg/mL) in an aqueous vehicle as solid-state drug nanoparticles. The novel NS system provided rapid dissolution of the poorly water-soluble compound with a drastically increased surface area, achieving over 90% drug release within 10 min. Moreover, the AP-loaded NS system afforded higher initial skin accumulation compared to ME, exhibiting about 4.0 times higher skin deposition after 3 h post-application. Our findings suggest that the high-payload, carrier-free nanosystem can be an effective tool for the skin delivery of AP. We are now planning to conduct preclinical local tolerability and efficacy studies, a physicochemical stability test, and even scale up the study for further clinical application of the high-payload AP NS system.

## Figures and Tables

**Figure 1 pharmaceutics-16-00171-f001:**
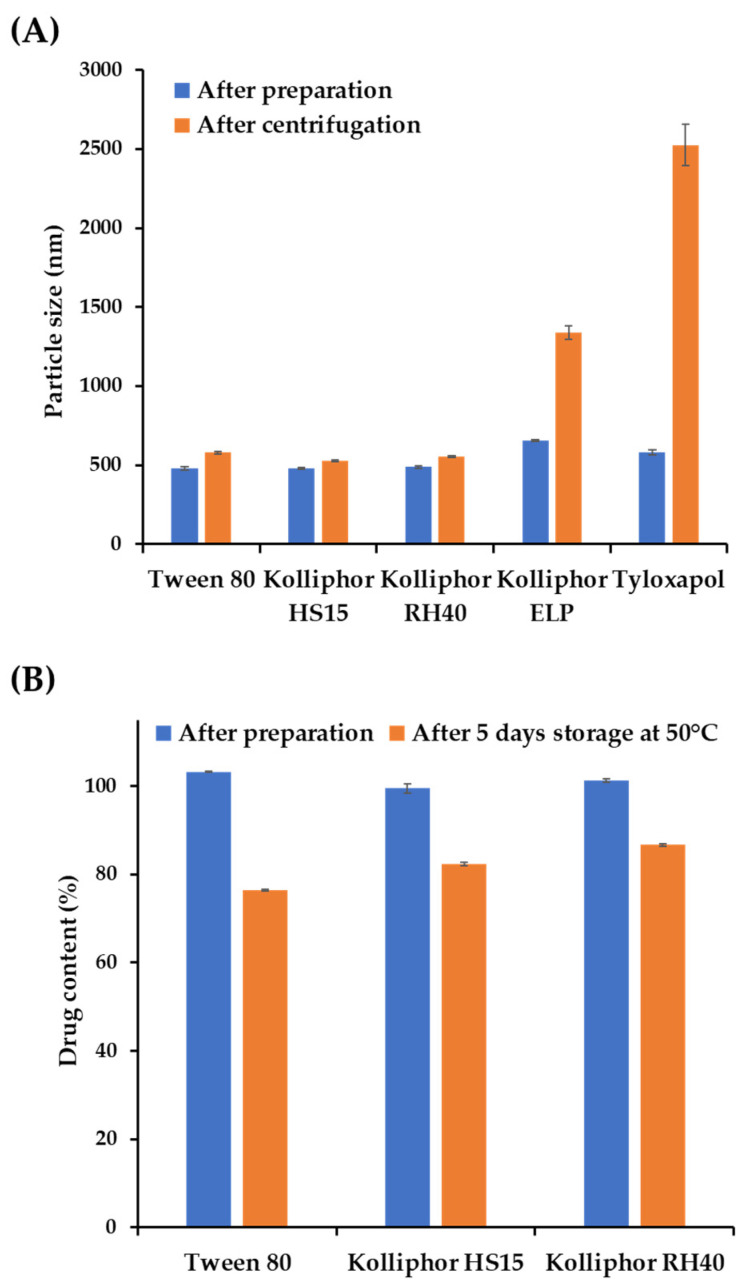
Physicochemical stability of the nanosuspensions (NSs) with different suspending agents under stress conditions. (**A**) Particle size and homogeneity of ascorbyl palmitate (AP) NSs with different suspending agents immediately after preparation and after centrifugation at 13,000 rpm for 10 min. (**B**) Changes in the AP content of NSs prepared with different suspending agents after five days of storage at 50 °C. Notes: The suspending agent concentration in the vehicle was set to 1 *w*/*v*% with a milling speed of 1500 rpm. Data are represented as the mean ± standard deviation (SD).

**Figure 2 pharmaceutics-16-00171-f002:**
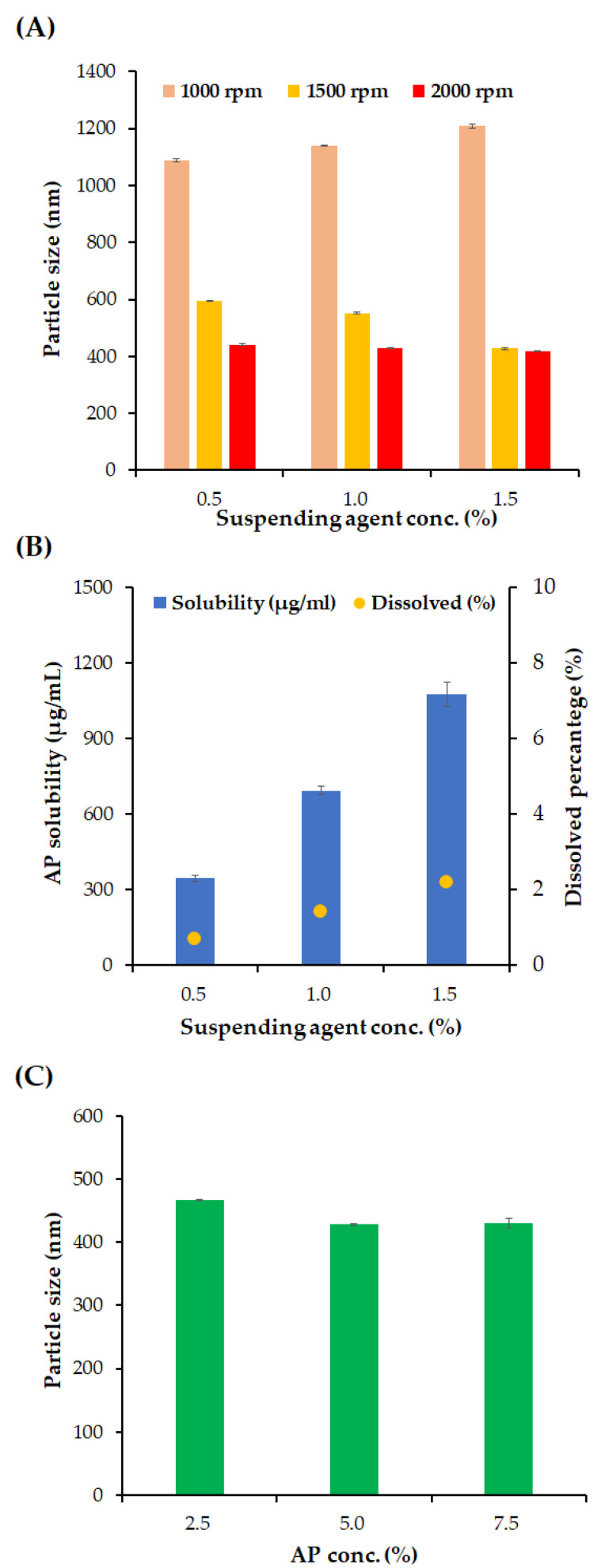
Effects of milling speed, concentration of suspending agents, and AP content on the particle size of NS and dissolved amount of AP. (**A**) Effects of suspending agent concentration (Kolliphor^®^ RH40, 0.5–1.5 *w*/*v*%) and milling speed (500–1500 rpm) on the particle size of NS. (**B**) Effects of suspending agent concentration (Kolliphor^®^ RH40, 0.5–1.5 *w*/*v*%) on AP solubility (μg/mL) and dissolved percentage (%) of AP in NS. (**C**) Effect of drug concentration (AP, 2.5–7.5 *w*/*v*%) on the particle size of NS. Notes: (**A**) Concentration of AP in the vehicle was fixed to 50 mg/mL, and the milling time was set to 1 h. (**B**) Concentration of AP in the vehicle was fixed to 50 mg/mL, and milling was performed at 2000 rpm for 1 h. (**C**) Concentration of the suspending agent was set to 0.5 *w*/*v*%, and milling was performed at 2000 rpm for 1 h. Data are represented as the mean ± SD.

**Figure 3 pharmaceutics-16-00171-f003:**
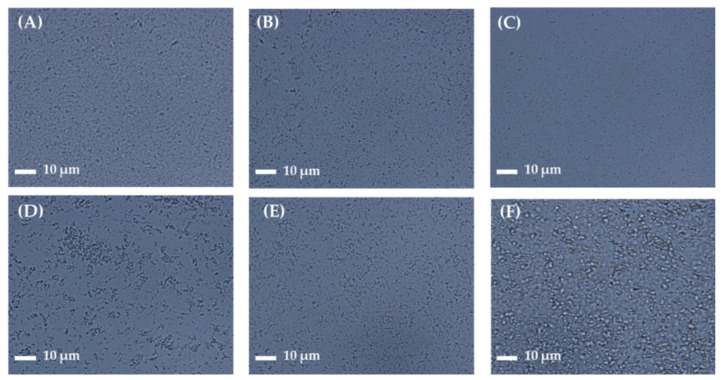
Microscopic observation of AP NS gel (NS-G) formulations with different thickening agents. Representative microscopic images of (**A**) NS (without thickening agent), NS-Gs with (**B**) Carbopol^®^ (1 *w*/*v*%), (**C**) sodium carboxymethyl cellulose (Na.CMC 700 K; 1 *w*/*v*%), (**D**) hydroxyethyl cellulose (HEC; 5 *w*/*v*%), (**E**) hydroxypropylmethyl cellulose (HPMC; 2 *w*/*v*%), and (**F**) xanthan gum (1 *w*/*v*%).

**Figure 4 pharmaceutics-16-00171-f004:**
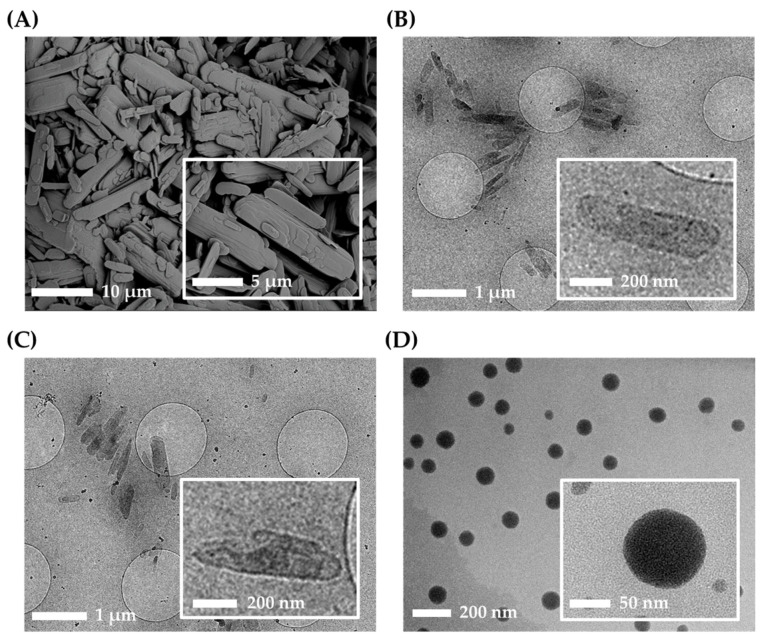
Morphological characteristics of AP and nanoformulations. (**A**) Scanning electron microscope (SEM) image of AP raw material. CryoTEM images of (**B**) NS, (**C**) NS-G, and (**D**) microemulsion gel (ME-G).

**Figure 5 pharmaceutics-16-00171-f005:**
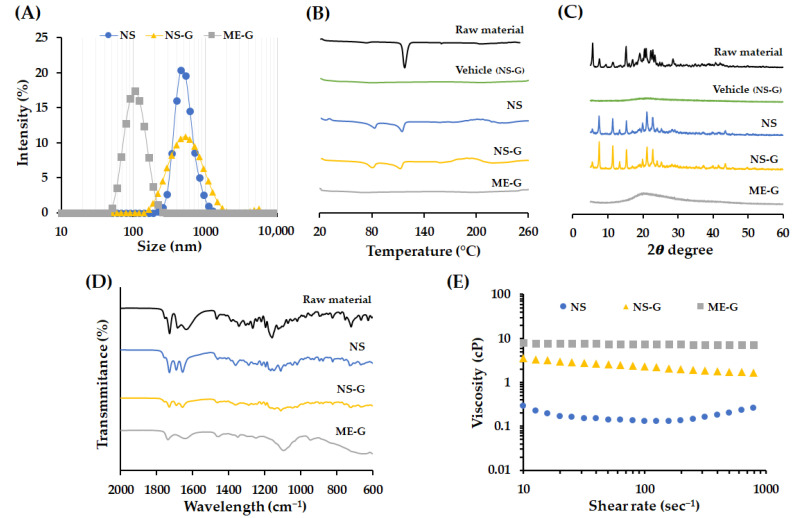
Physicochemical characteristics of AP and nanoformulations. (**A**) Size distribution of NS, NS-G, and ME-G. (**B**) Differential scanning calorimetry (DSC) thermogram and (**C**) X-ray diffraction (XRD) patterns of raw material, vehicle, NS, NS-G and ME-G. (**D**) FTIR spectra of raw material, NS, NS-G, and ME-G. (**E**) Representative viscosity curve of the delivery vehicle and AP formulation at the shear rate of 10–1000 s^−1^.

**Figure 6 pharmaceutics-16-00171-f006:**
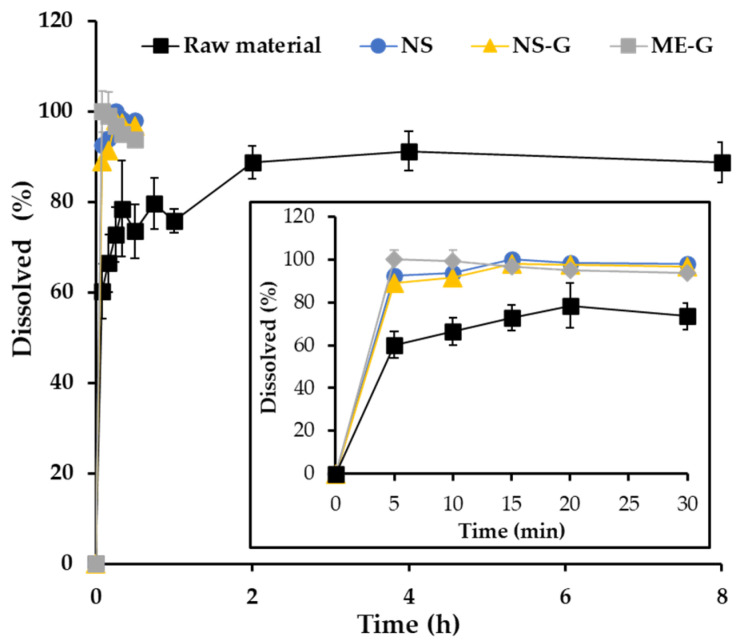
In vitro dissolution profiles of AP in the raw material (■), NS (●), NS-G (▲), and ME-G (■). Note: Data are represented as the mean ± SD (*n* = 3).

**Figure 7 pharmaceutics-16-00171-f007:**
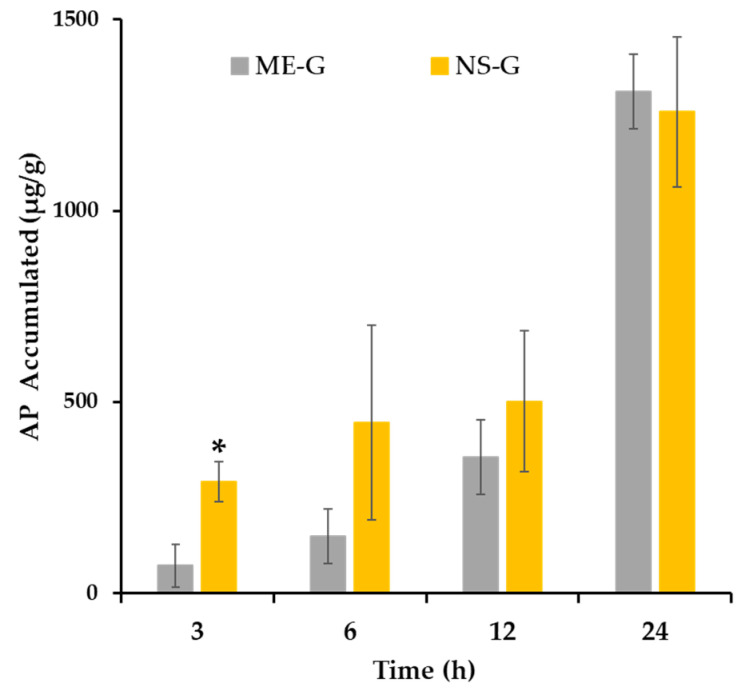
Ex vivo AP accumulation profile after the application of ME-G and NS-G via pig dorsal skin. Notes: Data represented as the mean ± SD (*n* = 3–5). * A significant difference between NS-G and ME-G at the same time point (*p* < 0.05).

**Table 1 pharmaceutics-16-00171-t001:** Effect of the suspending agent on the appearance, particle size and homogeneity of AP suspension.

Stabilizer ^a^	Appearance	Particle Size (nm) ^b,c^	Homogeneity ^b,d^
Surfactants			
Tween 80	Flowable, homogeneous	479.70 ± 8.50	0.31 ± 0.04
Kolliphor^®^ HS15	Flowable, homogeneous	479.47 ± 4.26	0.21 ± 0.02
Kolliphor^®^ RH40	Flowable, homogeneous	488.13 ± 7.21	0.26 ± 0.02
Kolliphor^®^ ELP	Flowable, homogeneous	655.63 ± 5.42	0.25 ± 0.01
Tyloxapol	Viscous, homogeneous	582.07 ± 14.9	0.21 ± 0.05
Poloxamer^®^ 188	Flowable, homogeneous	534.75 ± 2.65	0.28 ± 0.01
Hydrophilic polymers			
PEG 4000	Viscous, homogeneous	2856.3 ± 95.6	0.43 ± 0.06
PVP K17	Viscous, homogeneous	771.8 ± 11.0	0.22 ± 0.01
Na.CMC 90 K	Viscous, homogeneous	2326.7 ± 60.3	0.04 ± 0.01

^a^ The concentration of AP and stabilizer in the vehicle were fixed to 50 and 10 mg/mL, respectively. ^b^ Data expressed as mean ± SD (*n* = 3). ^c^ Indicates the mean hydrodynamic size determined using dynamic light scattering measurement technology. ^d^ Polydispersity index, calculated by dividing the square of the standard deviation by the square of the mean particle diameter.

**Table 2 pharmaceutics-16-00171-t002:** Physicochemical characteristics of AP NS and AP-loaded NS-G, ME-G.

Parameters	NS	NS-G	ME-G
AP conc. (mg/mL) ^a^	49.69 ± 0.01	49.74 ± 0.01	52.75 ± 0.03
Suspended (mg/mL)	49.34 ± 0.01	49.41 ± 0.01	N.A. ^e^
Dissolved (mg/mL)	0.35 ± 0.01	0.33 ± 0.01	52.75 ± 0.03
Particle size (nm) ^a^	489.5 ± 2.6 ^b^	493.2 ± 2.3 ^b^	103.1 ± 1.7 ^b^
Homogeneity ^a^	0.11 ± 0.03 ^c^	0.11 ± 0.03 ^c^	0.272 ± 0.03 ^c^
Zeta potential (mV) ^a^	−33.0 ± 0.7 ^a^	−48.7 ± 0.5 ^a^	−41.0 ± 4.9 ^a^
pH	4.5 ± 0.1	4.5 ± 0.1	4.4 ± 0.1
Viscosity (cP) ^d^	0.13	2.30	7.43

^a^ Data expressed as mean ± SD (*n* = 3). ^b^ Indicates the mean hydrodynamic size determined using dynamic light scattering measurement technology (Zetasizer Nano instruments). ^c^ Polydispersity index, calculated by dividing the square of the standard deviation by the square of the mean particle diameter. ^d^ Determined by a rotational rheometer at 25 °C with a shear rate 100 s^−1^. ^e^ Not available.

## Data Availability

The data presented in this study are available in the main text of this article.

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
