# Peer review of "Design of High-Payload Ascorbyl Palmitate Nanosuspensions for Enhanced Skin Delivery"

_pharmaceutics, 2024, doi:10.3390/pharmaceutics16020171_

Round 1

Reviewer 1 Report

Comments and Suggestions for Authors

Manuscript “Design of High-Payload Ascorbyl Palmitate Nanosuspensions for Enhanced Skin Delivery” represents a contribution to field of fundamental and applied research in Pharmaceutics Sciences. Text is clear and easy to read. The research topic is relative original. The research topic presented in the manuscript is current. The literature used is relative adequate. In order to increase the quality of manuscripts and in accordance with the quality of the Pharmaceutics (IF 5.4), before accepting the manuscript, it is essential that the authors (make corrections):

  • Add references in the Introduction: https://doi.org/10.3390/app10051594 https://doi.org/10.1016/j.ijpharm.2007.11.062
  • The results of FTIR analysis should be added to the existing results. It is necessary to present FTIR spectra of AP, AP-NS-G and AP-ME-G. Discuss the obtained results, please. Did the AP structure change during the preparation of the micro/nano emulsion?
  • Add DSC of ME-G in Fig. 5B., and XRD of ME-G  in Fig. 5C. Discuss the obtained results, please.

Reviewer 2 Report

Comments and Suggestions for Authors

The manuscript describes a sophisticated formulation for transdermal application of active molecules. The topic is interesting both for pharmaceutical and cosmetic applications (the later not mentioned in the text). The manuscript is clearly writen, but changes are necessary before it can be accepted for publication

COMMENTS:

1.- The manuscript should compare the results obtained with other previous formulations, both gels, as in this work, or liposomes, etc. This is important for non expert readers, as well as for industrial formulators.

2.- The authors should include some comments justifying that the performance on back pig skin (stratum corneum) can be translated to that on human skin.

3.- I would suggest to write a short appendix describing the method for measuring the transdermal permeability. This might be unknown for most people.

4.- Can the authors make a comparison of the economic issues related to the commercialization of this formulation with currently available ones?

5.- Comments on the environmental issues, e.g. degradability, appearence in waste water after washing, origin of components from natural sources, etc. 

Reviewer 3 Report

Comments and Suggestions for Authors

The MS "Design of High-Payload Ascorbyl Palmitate Nanosuspensions for Enhanced Skin Delivery" by Ho et.al., addresses an important issue of improving skin delivery of the antioxidant ascorbyl palmitate (AP) using a nanosuspension (NS) approach. Developing effective topical antioxidant formulas is valuable. The authors systematically optimized key parameters like milling intensity, stabilizer concentration, etc. to fabricate optimal AP nanosuspensions. The quality-by-design approach is rigorous.  A range of characterization techniques like SEM, TEM, DSC, etc. provided insights into NS properties. The NS showed good long-term chemical and physical stability.

Rapid in vitro drug dissolution was demonstrated for the NS compared to raw AP powder. This is beneficial for skin permeation. Higher AP skin accumulation was achieved with the NS gel vs. microemulsion gel in ex vivo studies. The effects are well explained.

 Minor comments:

a)       While the NS fabrication method and short-term stability data are thorough, long-term physical and chemical stability testing over months could further support product robustness.

b)      The statistical analysis is missing p values and indicators of significance for some key skin permeation data comparisons between NS and microemulsion gels.

c)       Careful proofreading is needed.

d)      The conclusion could include potential future work like preclinical tests, scaled manufacturing studies, etc. to translate this fundamental NS research further.

 Overall, the paper makes a meaningful contribution in designing and testing AP nanosuspensions for improved skin delivery. Addressing the suggestions above would further strengthen the quality and impact of the work.

Round 2

Reviewer 1 Report

Comments and Suggestions for Authors

The authors corrected the manuscript in accordance with the suggestions of the reviewer.

Reviewer 2 Report

Comments and Suggestions for Authors

The authors have modified the original manuscript to include the comments and suggestions made to the original one. Therefore, this revised version can be accepted for publication.